# Dynamic allosteric networks drive adenosine A₁ receptor activation and G-protein coupling

**Miguel A Maria-Solano\*, Sun Choi\***

Global AI Drug Discovery Center, College of Pharmacy and Graduate School of Pharmaceutical Science, Ewha Womans University, Seoul, Republic of Korea

**Abstract** G-protein coupled receptors (GPCRs) present specific activation pathways and signaling among receptor subtypes. Hence, an extensive knowledge of the structural dynamics of the receptor is critical for the development of therapeutics. Here, we target the adenosine A₁ receptor (A₁R), for which a negligible number of drugs have been approved. We combine molecular dynamics simulations, enhanced sampling techniques, network theory, and pocket detection to decipher the activation pathway of A₁R, decode the allosteric networks, and identify transient pockets. The A₁R activation pathway reveals hidden intermediate and pre-active states together with the inactive and fully-active states observed experimentally. The protein energy networks computed throughout these conformational states successfully unravel the extra and intracellular allosteric centers and the communication pathways that couple them. We observe that the allosteric networks are dynamic, being increased along activation and fine-tuned in the presence of the trimeric G-proteins. Overlap of transient pockets and energy networks uncovers how the allosteric coupling between pockets and distinct functional regions of the receptor is altered along activation. Through an in-depth analysis of the bridge between the activation pathway, energy networks, and transient pockets, we provide a further understanding of A₁R. This information can be useful to ease the design of allosteric modulators for A₁R.

**\*For correspondence:**
biochem0904@gmail.com
(MAM-S);
sunchoi@ewha.ac.kr (SC)

**Competing interest:** The authors declare that no competing interests exist.

## eLife assessment

The authors describe the dynamics underlying allostery of the adenosine A1 receptor, providing **valuable** insights into the receptor's activation pathway. The enhanced sampling molecular dynamics simulations of available structural data, followed by network analysis, reveal transient conformational states and communication between functional regions. The authors carefully state the limitations of their work, including the restricted convergence of the free energy landscape and missing water-mediated hydrogen bond coordination. Collectively, they provide a **convincing** framework for advancing rational design strategies of specific modulators with desired modes of action.

[Editors' note: this was originally reviewed and assessed by Biophysics Colab]

## Introduction

Biomolecules present intrinsic dynamism and plasticity to respond to physiological changes (*Wodak et al., 2019*). In transmembrane receptors, the binding of ligands in the orthosteric site located at the extracellular region can establish dynamic and chemical communication with the intracellular region resulting in key structural rearrangements that trigger a particular response (*Nussinov et al., 2014*). This communication between distant protein sites is called allostery and its understanding presents a current challenge for many protein complexes (*Wodak et al., 2019*; *Tsai and Nussinov, 2014*;

*Nussinov and Tsai, 2015*; *Guo and Zhou, 2016*). Allosteric modulators are capable to bind regions other than orthosteric sites and propagate communication networks to other functional regions of the receptor (*Dokholyan, 2016*). A complete information for the structure-based design of allosteric modulators involves not only the exploration of the receptor activation pathway and the identification of binding sites, (*Lu et al., 2021*) but also the characterization of the allosteric pathways, which is a complicated task. However, allosteric drugs offer the advantage of higher selectivity with respect to conventional drugs due to the greater sequence variability of allosteric sites. (*May et al., 2007*; *Christopoulos, 2002*).

In G-protein coupled receptors (GPCRs), the binding of endogenous agonists in the orthosteric site reshapes the conformational ensemble of a Transmembrane (TM) helix to prime the binding and activation of G-proteins. (*Erlandson et al., 2018*; *Weis and Kobilka, 2018*; *Mafi et al., 2022*) The conformational landscapes of GPCRs associated with receptor activation are complex, with some general principles in common and also receptor-specific differences. (*Bostock et al., 2019*; *Zhou et al., 2019*) For many GPCRs it has been observed that (1) inactive and pre-active states are in dynamic equilibrium in the apo form, (2) the action of the agonist induces subtle population shifts involving the stabilization of intermediate and pre-active states and/or decreasing their rates of interconversion and (3) the combined action of both, agonist and G-protein binding stabilizes a more rigid fully-active state. (*Weis and Kobilka, 2018*; *Bostock et al., 2019*; *Mattedi et al., 2020*; *Miao and McCammon, 2016*; *Fleetwood et al., 2021*) The complexity of GPCRs activation landscapes hinders its experimental and computational characterization. However, the study of its system-specific properties opens the door for the design of allosteric drugs.

The adenosine $A_1$ receptor ($A_1$R) is a member of the class A G protein-coupled receptors (GPCRs) family that preferentially couples with Gi/o proteins. It is widely distributed in multiple organs mediating a variety of physiological processes, including those in the brain and the heart. Thus, $A_1$R has significant therapeutic potential in the treatment of numerous diseases and disorders. (*Deb et al., 2019*) In fact, it has been targeted for pain management through allosteric modulation although without success in clinical trials. (*Draper-Joyce et al., 2021*; *Bruns and Fergus, 1990*) Fortunately, X-ray crystallography and Cryo-electron microscopy (Cryo-EM) captured $A_1$R in inactive and active states revealing a notorious inward-to-outward conformational transition of TM6 (*Figure 1A* and *Figure 1—figure supplement 1*). (*Draper-Joyce et al., 2021*; *Draper-Joyce et al., 2018*; *Glukhova et al., 2017*) The most recently released active state structure is resolved with adenosine, trimeric G-protein, and also with a positive allosteric modulator (PAM) located in an extrahelical region. (*Draper-Joyce et al., 2021*) Computational and mutagenesis studies have reported interesting insights about the $A_1$R activation and allosteric modulation identifying important residues for the signaling efficacy of agonists and cooperativity of PAMs. (*Draper-Joyce et al., 2021*; *Nguyen et al., 2016b*; *Do et al., 2022*) Despite all these achievements, a detailed characterization of the allosteric networks that drive receptor activation and G-protein binding is still missing.

In this work, we develop a computational workflow tailored to decipher the interplay between the receptor activation pathway, the allosteric communication networks, and transient pockets. First, we use conventional molecular dynamics (MD) simulations coupled with enhanced sampling techniques to reconstruct the receptor activation conformational landscape of the inward-to-outward TM6 transition revealing the inactive, intermediate, pre-active, and fully-active conformational states. Second, we study the dynamic communication energy networks throughout the conformational states sampled successfully capturing the extra and intracellular communication centers and the pathways that interconnect them. We observe that the allosteric communication is enhanced along the receptor activation and fine-tuned in the presence of the trimeric G-protein. Third, we use a geometry-based approach to search for the formation of transient pockets. By studying the connection between these three elements we give a complete dynamic picture of the $A_1$R activation that is essential for the design of specific allosteric modulators. This in silico approach can be also applied to uncover the activation landscape, allosteric networks, and transient pockets of related GPCRs, which is of interest for allosteric drug design.

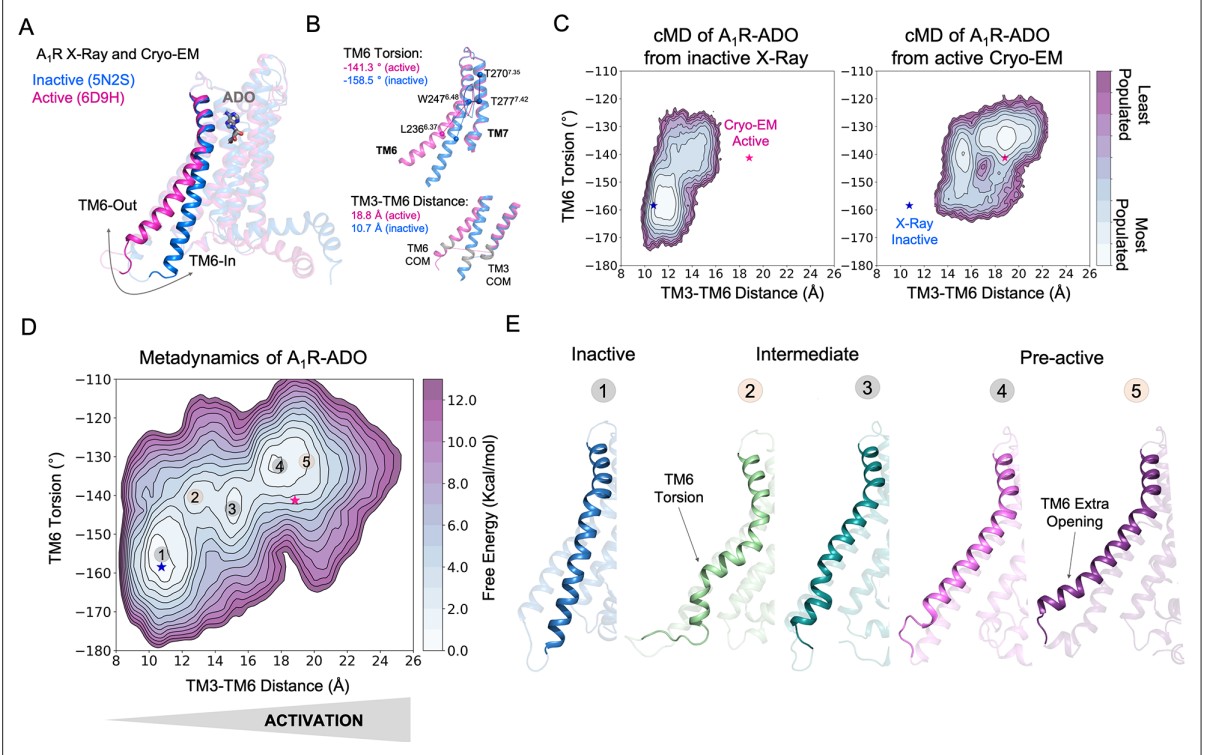

**Figure 1.** Free energy landscape (FEL) of A₁R activation in the presence of adenosine (ADO). (**A**) TM6 inward-to-outward conformational transition observed in the inactive (PDB 5N2S) and active (PDB 6D9H) X-Ray and Cryo-EM structures. (**B**) Features used to follow the TM6 inward-to-outward transition (receptor activation) in this work. The TM6 torsion corresponds to the dihedral angle formed by the alpha carbon atoms of L236$^{6.37}$, W247$^{6.48}$, T277$^{7.42}$, and T270$^{7.35}$. For the TM3-TM6 intracellular ends distance, we computed the center of mass (COM) distance between the backbone atoms of TM3(Y226$^{6.27}$, G227$^{6.28}$, K228$^{6.29}$, L230$^{6.30}$, and E230$^{6.31}$) and TM6(R105$^{3.50}$, Y106$^{3.51}$, L107$^{3.52}$, R108$^{3.53}$, and V109$^{3.54}$). The X-ray and Cryo-EM values are also shown. (**C**) Population analysis obtained from conventional molecular dynamics (cMD) simulations of A₁R activation starting from the inactive X-Ray and active Cryo-EM structures, the inactive X-ray and active Cryo-EM coordinates are projected as blue and magenta stars, respectively. (**D**) Reconstruction of the FEL associated with the A₁R activation obtained from metadynamics simulations. The most relevant conformational states are labeled from 1 to 5. Note that the lowest energy states (1,3 and 4) are labeled in gray while the others (2 and 5) are in orange. The X-ray and Cryo-EM coordinates are also projected. (**E**) Representative structures of the inactive (1), Intermediate (2-3), and Pre-active (4-5) conformational states sampled along A₁R-ADO activation.

The online version of this article includes the following figure supplement(s) for figure 1:

**Figure supplement 1.** Representation of the A₁R receptor in complex with heterotrimeric G$_{i2}$ protein (PDB 6D9H).

**Figure supplement 2.** Structures used as starting points for the walker metadynamics simulations.

**Figure supplement 3.** Estimate of the free energy differences between the energy minima of the free energy surface.

**Figure supplement 4.** Evolution of the CV1 (TM3-TM6 Distance) over the simulation time.

**Figure supplement 5.** 2D free energy landscape of A₁R associated with the TM3-TM6 intracellular ends distance and its associated error.

**Figure supplement 6.** Representation of relevant micro-switches for A₁R.

**Figure supplement 7.** Reweighting of the metadynamics simulations onto 2D free energy profiles.

**Figure supplement 8.** Reweighting of the metadynamics simulations onto 3D free energy profiles.

## Results

### The free energy landscape of A₁R activation reveals intermediate and pre-active states

To uncover the activation pathway of A₁R in the presence of its endogenous agonist adenosine (A₁R-ADO), we reconstructed the free energy landscape (FEL) of A₁R-ADO associated with the TM6 inward-to-outward transition observed by X-ray and Cryo-EM data (**Figure 1A**). Specifically, we focus on two features: the TM6 torsion and the center of mass (COM) distance between the TM3 and TM6 intracellular ends (**Figure 1B**). Initially, we performed conventional molecular dynamics (cMD)

simulations starting from both, the inactive (PDB 5N2S) and the active (PDB 6D9H) structures in order to increase the sampling of the activation conformational space. For each starting point, we computed three replicas of 500ns, which is a reasonable simulation time to provide an initial sampling of the receptor activation. Either starting from the inactive or active structures the complete TM6 Inward-to-Outward transition is not sampled (*Figure 1C*). However, when starting from the active structure, an intermediate state centered at ca. TM6 torsion (–140°) and TM3-TM6 distance (15 Å) coordinates is substantially populated. This intermediate structure still presents a rather high degree of TM6 torsion while the TM3-TM6 distance is shortened. In contrast, when starting from the inactive structure, the receptor mostly samples inactive conformations exhibiting a low probability of progressing toward the intermediate state, thus suggesting a higher inactive-to-intermediate transition time scale. For a better characterization of the conformational landscape of the $A_1R$-ADO activation and to check the importance of the intermediate state, we relied on metadynamics simulations. The primary objective of this calculation is to identify and reconstruct the major conformational states that are involved in the pathway of receptor activation. Metadynamics is a powerful method that has been successfully used to study complex conformational transitions in proteins (*Calvó-Tusell et al., 2022*; *Maria-Solano et al., 2019*; *Kuzmanic et al., 2017*), including GPCRs (*Mafi et al., 2022* ; *Mattedi et al., 2020*). In particular, we performed well-tempered metadynamics (WT-MetaD) simulations using the walkers approach (see Materials and methods). We selected 10 representative structures along the activation pathway sampled in the cMD as starting points for the walker replicas (see *Figure 1—figure supplement 2*). After 250 ns of accumulated time, we successfully reconstructed the major conformational states of the FEL (see convergence assessment in *Figure 1—figure supplements 3–4* and error estimation in *Figure 1—figure supplement 5*).

The FEL in the presence of adenosine confirms that $A_1R$ presents three major states in dynamic equilibrium (inactive, intermediate, and pre-active) while the fully-active Cryo-EM like state is not significantly populated. The relative stabilities of the three states show that the most stable state is the inactive, followed by the pre-active and intermediate states, which are slightly higher in energy. The inactive state resembles the inactive X-ray coordinates showing short TM3-TM6 distances and low TM6 torsion values (state 1 in *Figure 1D–E*). The inactive conformations are stabilized by a tight energy coupling between TM3 and TM6 (see below). As the activation progresses, TM6 torsion evolves further than the TM3-TM6 distance. At this point, the remaining energy coupling between TM3 and TM6 ends hampers the progression of the TM6 outward transition. This tug of war between forces provokes the adoption of torsion in the TM6 end that is signature of the intermediate conformations (state 2 in *Figure 1D–E*). This notorious torsion evolves to a more stable intermediate conformation (state 3 in *Figure 1D–E*). Finally, a pre-active local energy minimum is reached completing the TM6 transition, which involves the complete break between TM6 and TM3 energy coupling (see below). The pre-active state presents similar TM3-TM6 distances to the fully-active Cryo-EM structure. However, it exhibits higher TM6 torsion values (state 4 in *Figure 1D–E*). In addition, an extra opening of TM6 is accessible (state 5 in *Figure 1D–E*). All data together suggests that the coupling of the G-proteins is required to stabilize the adoption of the fully-active Cryo-EM like structure (see next sections). As a complementary analysis, we conducted the reweighting of the metadynamics simulations (*Branduardi et al., 2012*) to determine the free energy as a function of previously identified $A_1R$ micro-switches (ionic-lock, PIF motif, water-lock, and toggle switch). The fact that we capture the distinct energy barriers associated with unbiased micro-switches highlights the accuracy of the metadynamics simulations in reproducing the pathway of activation and provides useful information to guide the selection of collective variables for future GPCR landscape calculations (*Figure 1—figure supplements 6–8*). Once deciphered the $A_1R$-ADO FEL of activation, our study follows on investigating the receptor-specific allosteric properties that harbor these inactive, intermediate, and pre-active states.

## Energy networks capture the dynamic allosteric pathways along $A_1R$ activation

A complete understanding of allosteric modulation involves the decoding of the communication pathways that dynamically couple distinct protein sites. Despite the difficulties, network theory has been successfully applied to uncover the allosteric communication pathways in protein complexes (*Calvó-Tusell et al., 2022*; *Maria-Solano et al., 2021*), including GPCRs (*Kong and Karplus, 2007*; *Lee et al., 2014*). In order to trace down the allosteric pathways in $A_1R$-ADO, we relied on the protein

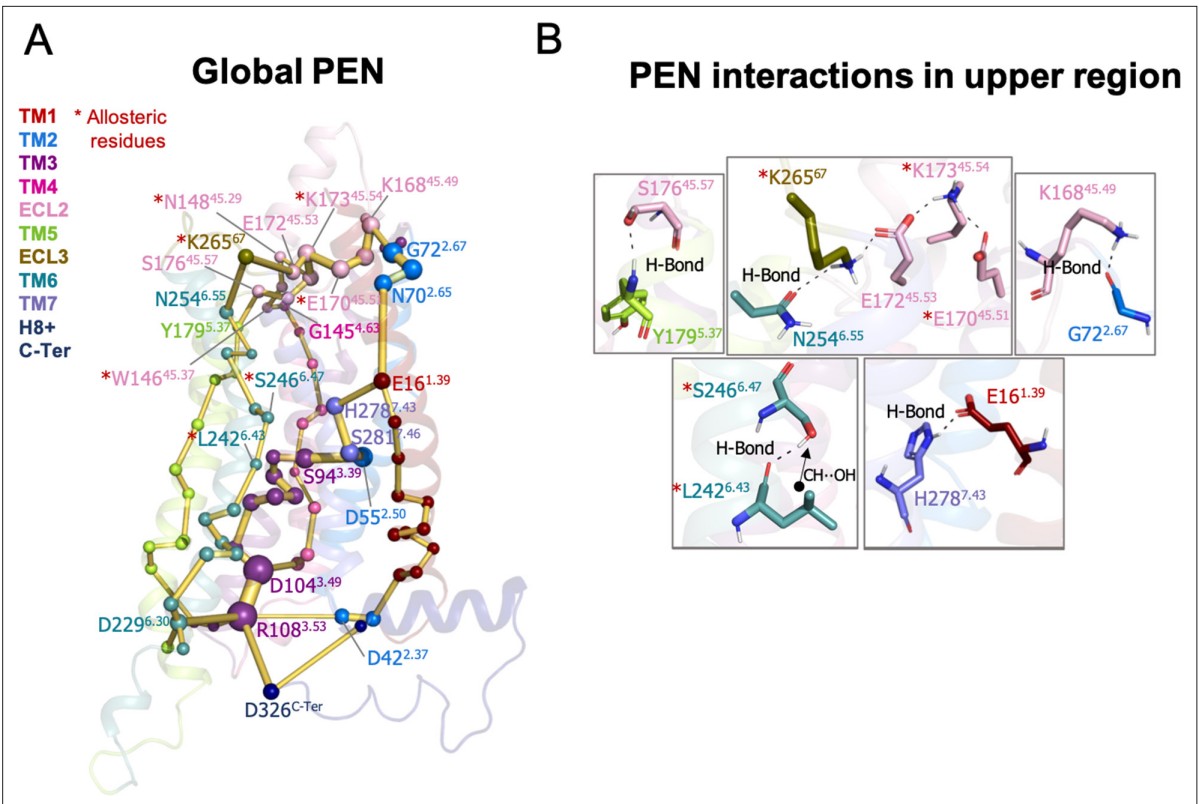

**Figure 2.** Protein energy networks (PEN) of A$_1$R-ADO conformational ensemble. The PEN identifies extra and intracellular communication centers together with the allosteric pathways that interconnect them. The PEN residues (nodes) are represented by colored spheres as a function of the receptor region (e.g. TM6 nodes in teal) while the allosteric pathways (edges) as yellow-orange sticks. The size of each edge and node corresponds to their importance for allosteric communication. The experimentally identified allosteric residues captured in the PEN nodes are labeled with a red asterisk. (**B**) Relevant interactions of the PEN in the upper region of the receptor.

The online version of this article includes the following figure supplement(s) for figure 2:

**Figure supplement 1.** Adenosine interactions with protein energy networks (PEN) residues in the A$_1$R-ADO conformational ensemble.

energy networks (PEN) approach (*Serçinoğlu and Ozbek, 2018*). First, we used the get Residue Interaction eNergies and Networks (gRINN) (*Serçinoğlu and Ozbek, 2018*) tool to calculate the pairwise residue interaction energies along the A$_1$R-ADO conformational ensemble and obtain the mean interaction energy matrix (IEM). Second, we processed the mean IEM into the shortest-path map (SPM; *Maria-Solano et al., 2021*; *Romero-Rivera et al., 2017*; *Osuna, 2021*) tool to construct and visualize the PEN graph.

The analysis of the PEN associated with the A$_1$R-ADO ensemble shows that the extracellular region can communicate with the intracellular region through multiple energy pathways (see *Figure 2A*). In the extracellular region, ECL2 and ECL3 present a center of communication connecting with TM4, TM5, TM1, and TM6. Energy pathways involving TM5, TM6, and TM4 propagate from the extracellular ECL2/ECL3 regions and link with TM3 in the intracellular region. In addition, TM3 establishes connections with TM2 and TM7 that communicate back to the ECL2 by crosslinking with TM1 (*Figure 2A–B*). TM6 is then energetically coupled with the extracellular region through ECL3/ECL2 and with the intracellular region through TM3. In fact, the TM3 end plays a central role in allosteric communication at the intracellular region. Specifically, R108$^{3.53}$ is a communication hub forming and hydrogen networking with D229$^{6.30}$, D104$^{3.49}$ (ionic-lock micro-switch) and D326$^{C-Ter}$. Regarding the energy communication at the orthosteric site, adenosine samples a wide range of poses in its large binding site (*Draper-Joyce et al., 2021*; *Miao et al., 2018*) performing transient interactions with many residues of the PEN, thus establishing multiple transient energy communication that propagates towards the TM6 intracellular end (see *Figure 2—figure supplement 1*).

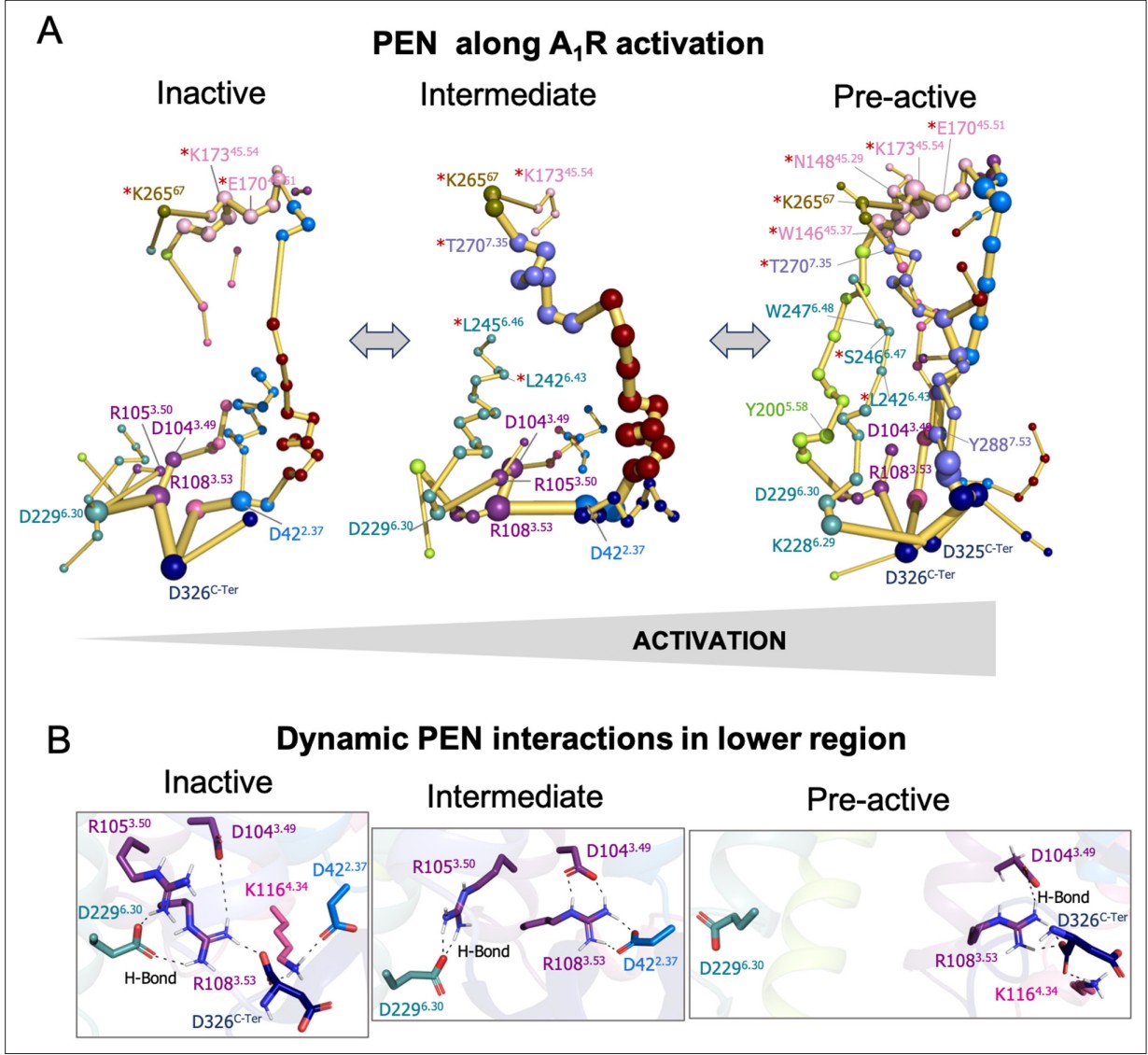

**Figure 3.** Protein energy networks (PEN) of A$_1$R-ADO in the inactive, intermediate, and pre-active states. (**A**) The PEN residues (nodes) are represented by colored spheres as a function of the receptor region (e.g. TM6 nodes in teal) while the allosteric pathways (edges) as yellow-orange sticks. The size of each edge and node corresponds to their importance for allosteric communication. The experimentally identified allosteric residues captured in the PEN nodes are labeled with a red asterisk. The allosteric communication is enhanced along the receptor activation. (**B**) Relevant interactions found in the PEN of the lower region of the receptor that are altered along activation.

The online version of this article includes the following figure supplement(s) for figure 3:

**Figure supplement 1.** Representation of the free energy landscape (FEL) of A$_1$R-ADO activation split into conformational states.

**Figure supplement 2.** Illustration of the conformational dynamics of the micro-switches along A$_1$R activation.

In order to provide further insights into the allosteric communication along the activation pathway, we decided to split the analysis into the inactive, intermediate, and pre-active states (*Figure 3* and *Figure 3—figure supplement 1*). We started the analysis with the inactive PEN, which shows that the extracellular ECL2 center is only connected to the intracellular region through TM2 and TM1. In the intracellular region, R108$^{3.53}$ establishes a tight communication with D104$^{3.49}$, D229$^{6.30}$, and D326$^{C-Ter}$, see *Figure 3A–B*. Note that D229$^{6.30}$ can also communicate with R105$^{3.50}$. This hydrogen-bonded network that captures the ionic-lock micro-switch is signature of the inactive in-ward conformation of TM6. In fact, D229$^{6.30}$-R108$^{3.53}$ interaction is pivotal to attaining the inactive conformation of the receptor. Interestingly D326$^{C-Ter}$ plays an important role in energy communication. The flexible C-Ter region can partially occupy the G-proteins binding site and perform stable communication

with R108$^{3.53}$. We followed the analysis with the PEN of the intermediate ensemble. In this case, the extracellular communication center is reduced and ECL2/ECL3 region is connected to the intracellular region through TM7 and TM1. Interestingly, the communication through TM6 starts to take place partially in the intracellular region. In contrast with the inactive ensemble, D229$^{6.30}$ now mainly communicates with R105$^{3.50}$ instead of R108$^{3.53}$ in the intracellular region, see *Figure 3A–B*. Therefore, at this point, the dynamics of the ionic-lock is altered. This transient interaction is key to preventing TM6 opening after the tension generated in the last segment of TM6 and provokes a torsion in the TM6 intracellular end that captures the receptor in an intermediate state (described above). In this scenario, R108$^{3.53}$ communicates with D42$^{2.37}$ and D104$^{3.49}$. Note that the C-Ter does not communicate with TM3, which explains the transience of this interaction due to the C-Ter flexibility. Finally, in the pre-active ensemble, the extracellular ECL2 networks are recovered and connected to the intrahelical region through multiple pathways, including TM2, TM4, TM5, TM6, and TM7. Interestingly, the PEN captures Y200$^{5.58}$ and Y288$^{7.53}$ in the TM5 and TM7 pathways, respectively. These tyrosine residues have been found to stabilize the active state through a hydrogen bond coordinated by a bridging water molecule in the so-called water-lock. However, the water-lock bridge is not observed because non-protein molecules are excluded in the PEN calculation, which points out a major limitation of the methodology. Regarding the TM6 pathway, its partial communication observed in the intermediate state progresses further capturing the toggle switch (W247$^{6.48}$) and it is completed linking with TM5 (*Figure 3A*). This enhanced communication between the intra and extracellular regions is driven by the complete opening of TM6. As expected, the TM6 opening provokes the break of the ionic-lock disconnecting TM6 (D229$^{6.30}$) from TM3 (R108$^{3.53}$ and R105$^{3.50}$). Indeed, the R108$^{3.53}$ node loses prominence in the intracellular region. However, R108$^{3.53}$ still can communicate with D326$^{C-Ter}$, which may compensate for the break of TM3-TM6 ends (*Figure 3A–B*). Complementary insights are gained by computing the histograms of relevant micro-switches along the inactive, intermediate, and pre-active states (*Figure 3—figure supplement 2*).

At this point, we wondered if the positions captured in the PEN coincide with allosteric residues previously identified in mutagenesis studies that affect the allosteric responses of the receptor. For the ECL2/ECL3 extracellular region, 9 allosteric residues have been identified that affect the efficacy of orthosteric agonists (*Nguyen et al., 2016b*; *Nguyen et al., 2016a*). Among all of them, up to 7 are captured in the ECL2/ECL3 PEN communication center (**W156$^{45.37}$**, **N148$^{45.29}$**, **K173$^{45.54}$**, **K265$^{67}$**, **E170$^{45.51}$**, **S150$^{45.31}$**, and **T270$^{7.35}$**) and the other 3 are located in the ECL2 alpha helix that connects with the PEN through S150. In addition, other four allosteric residues located in a TM1-7 extrahelical region have been recently reported to be important for PAM cooperativity (*Draper-Joyce et al., 2021*). The PEN directly captures **S246$^{6.47}$**, **L242$^{6.43}$**, and **L245$^{6.46}$**, and the other residue (**G279$^{7.44}$**) is

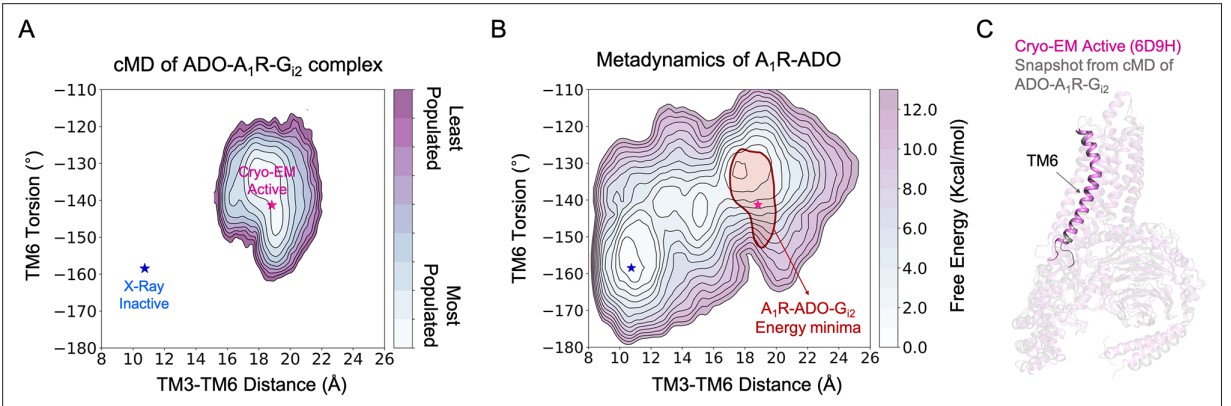

**Figure 4.** Effect of G-protein binding on the conformational ensemble of A$_1$R-ADO. (**A**) Population analysis of A$_1$R activation in the ADO-A$_1$R-G$_{i2}$ complex obtained from conventional molecular dynamics (cMD) simulations. The TM6 torsion corresponds to the dihedral angle formed by the alpha carbon atoms of L236$^{6.37}$, W247$^{6.48}$, T277$^{7.42}$, and T270$^{7.35}$. For the TM3-TM6 intracellular ends distance, we computed the center of mass (COM) distances between the backbone atoms of TM3(Y226$^{6.27}$, G227$^{6.28}$, K228$^{6.29}$, L230$^{6.30}$, and E230$^{6.31}$) and TM6(R105$^{3.50}$, Y106$^{3.51}$, L107$^{3.52}$, R108$^{3.53}$, and V109$^{3.54}$). The inactive and active X-ray and Cryo-EM coordinates are projected as blue and magenta stars, respectively. (**B**) Projection of the ADO-A$_1$R-G$_{i2}$ energy minima obtained from cMD over the FEL associated with the A$_1$R activation obtained from metadynamics simulations. The ADO-A$_1$R-G$_{i2}$ energy minima (depicted in red) is centered on the coordinates of the active Cryo-EM structure. (**C**) Overlay of the active Cryo-EM structure (PDB 6D9H) and a representative snapshot from the ADO-A$_1$R-G$_{i2}$ energy minima.

located adjacent to the PEN residue H278 and S246. Note that the experimentally identified allosteric residues captured in the PEN are labeled with a red asterisk in *Figure 2* and *Figure 3*. It is worth mentioning that among the allosteric residues captured in the PEN most of them are identified in the pre-active ensemble while only some of them are identified in the inactive and intermediate states. This is due to the enhanced allosteric communication observed upon activation. In summary, the PEN analysis along the conformational states captures many important positions for the allosteric mechanisms of the receptor and provides a dynamic view of the protein energy communication along the activation of the receptor. The TM6 in-ward to outward transition reveals that the TM6 allosteric pathway along the receptor is missing in the inactive state, partially formed in the intermediate state and completely established upon the receptor activation. Indeed, the pre-active state is characterized by increased allosteric communication through multiple pathways.

## G-protein binding stabilizes the fully-active state and fine-tunes the allosteric communication

The next unknown we were intrigued to investigate is how the binding of G-proteins affects the $A_1R$ conformational ensemble and the allosteric networks. To that end, we performed 3 cMD replicas of 500 ns of $A_1R$ in presence of both, adenosine and heterotrimeric $G_{i2}$ protein (referred as ADO-$A_1R$-$G_{i2}$ complex). As previously, the MD data was plotted as a function of the TM6 torsion and TM3-TM6 ends distance (*Figure 4A*). The population analysis of the receptor activation shows that the presence of the slim $G\alpha_{i2}$-$\alpha5$ helix in the $A_1R$ intracellular cavity prevents TM6 to sample intermediate and inactive conformations. Indeed, $A_1R$-$G_{i2}$ displays restrictive conformational dynamics by only sampling one major conformational state. This conformational state overlaps with both, the pre-active state and the fully-active state corresponding to the active Cryo-EM structure (*Figure 4B–C*). Thus, the binding of the G-proteins induces a population shift in $A_1R$ towards fully-active conformations, as described previously in other GPCRs (*Weis and Kobilka, 2018*; *Mafi et al., 2022*; *Bostock et al., 2019*; *Zhou et al., 2019*).

To study the allosteric pathways of the ADO-$A_1R$-$G_{i2}$ complex, we computed the PEN considering the intra ($A_1R$-$A_1R$) and inter ($A_1R$-$G\alpha_{i2}$) interactions. The PEN analysis shows some similarities with respect to the PEN of the pre-active ensemble (*Figure 5*). In both cases, TM6 allosteric communication is completed and TM7 presents an important communication pathway connecting the intra and extracellular regions. However, the presence of G-Protein finely tunes the PEN in some aspects: (1) the ECL2 extracellular communication center propagates toward the ECL2 helix capturing the **W156**[45.37] allosteric residue that was not identified in previous PEN analysis, (2) the communication pathways between extra and intracellular regions are refined only passing through TM2, TM1, TM6, and TM7 and not using TM4 and TM5, certainly TM7 becomes the major communication pathway and another allosteric residue **G279**[7.44] is captured as an important communication node, and (3) as expected, the C-Ter does not communicate with TM3 because the end of the $\alpha5$ helix of $G\alpha_{i2}$ ($G\alpha_{i2}$-$\alpha5$ helix) replaces the C-Ter communication position. As a consequence, TM3 (R108) communicates with TM2 (D42) and most importantly with $G\alpha_{i2}$-$\alpha5$ helix (D667). The latest becomes a communication hub in the intracellular region by also connecting with H8 (K294). The inclusion of $G\alpha_{i2}$ also causes TM6 (E229) to communicate with TM5 (R208), *Figure 5*. Thus, the receptor signaling attains a specific profile of protein energy communication networks triggered by the $G\alpha_{i2}$ protein coupling.

## Identification of transient pockets as potential allosteric sites

The identification of transient pockets formed along specific receptor activation profiles is useful to guide the design of allosteric drugs. We use a geometry-based algorithm (MDpocket; *Schmidtke et al., 2011*) in order to search for transient pockets in the inactive, intermediate, pre-active and fully-active ensembles of $A_1R$. The MDpocket output provides a normalized frequency map that allows the visualization of the frequency of a pocket formation along each conformational ensemble studied.

We identified multiple pockets in the upper and lower regions of the receptor. Interestingly, we observe that pockets can change their shape and frequency of formation along the different conformational states (*Figure 6A*). Focusing on the upper region pockets (**PA-D**), **PA** is present at the ECL2 vestibule with low frequency. This region has been proposed to bind $A_1R$ PAMs on the basis of computational and mutagenesis studies (*Nguyen et al., 2016b*; *Miao et al., 2018*; *Nguyen et al., 2016a*). **PB** is located at the adenosine binding site and extends to the adjacent secondary pocked observed

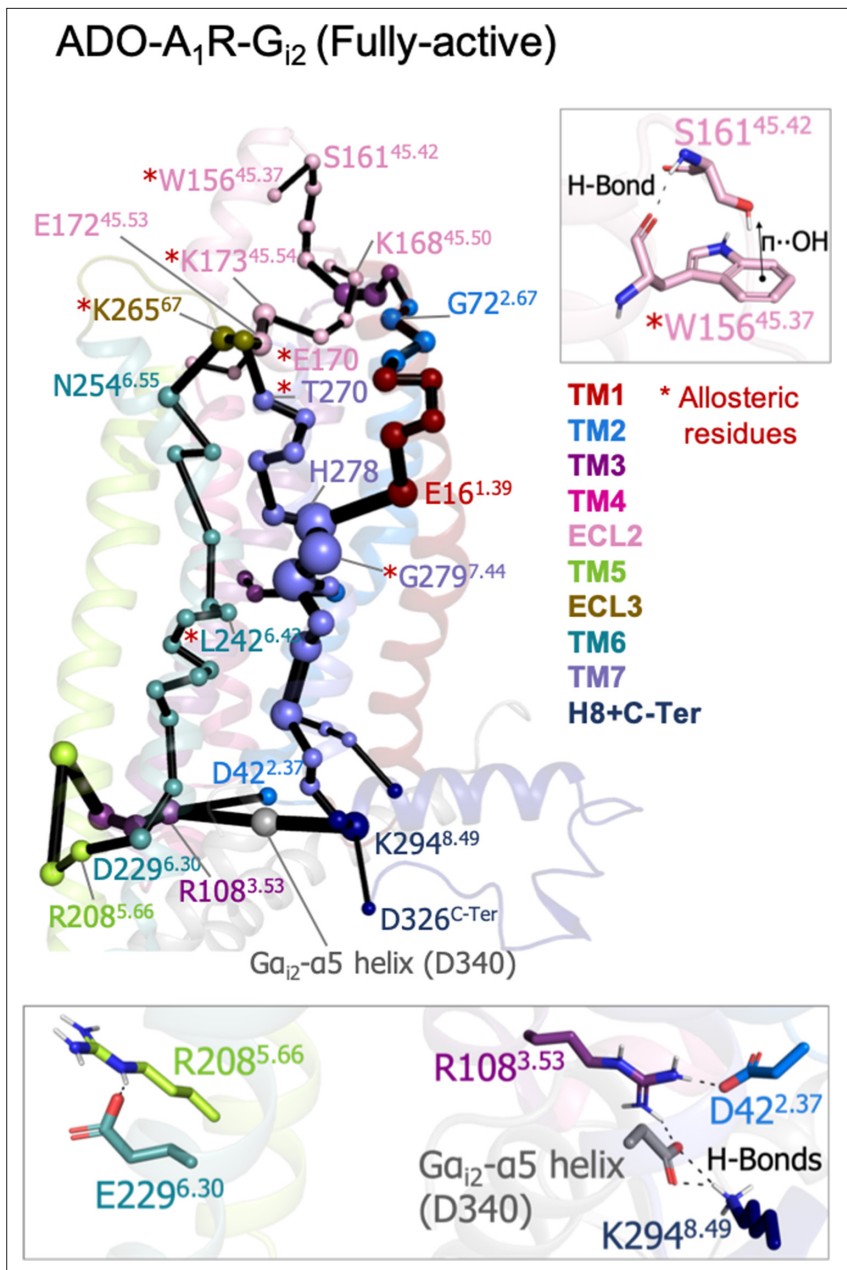

**Figure 5.** Effect of G-protein binding on $A_1R$-ADO the protein energy networks (PEN) of $A_1R$-ADO. The PEN residues (nodes) are represented by colored spheres as a function of the receptor region (e.g. TM6 nodes in teal) while the allosteric pathways (edges) as yellow-orange sticks. The size of each edge and node corresponds to their importance for allosteric communication. The experimentally identified allosteric residues captured in the PEN nodes are labeled with a red asterisk. Relevant interactions of the PEN in the upper and lower regions of the receptor are also shown.

by X-ray data in the inactive $A_1R$ structure (**Glukhova et al., 2017**). Due to the **PB** large size, it overlaps with many allosteric modulators observed in class A (PDBs 4MQT, **Kruse et al., 2013**) 5NDD, (**Cheng et al., 2017a**) 4MBS (**Tan et al., 2013**) and class C (PDBs 5CGC, **Christopher et al., 2015**) 5CGD, (**Christopher et al., 2015**) 6FFH (**Christopher et al., 2019**) and 6FFI (**Christopher et al., 2019**) GPCR structures. **PE** corresponds to the protein internal channel and overlaps with the NAM sodium ion site (PDB 4N6H), (**Fenalti et al., 2014**) which is narrowed along activation (**Figure 6A**). **PD** corresponds to the MIPS521 PAM shallow pocket exposed to the lipid-detergent interface (PDB 7LD3) (**Draper-Joyce et al., 2021**). To date, this is the only allosteric modulator whose structure has been resolved in

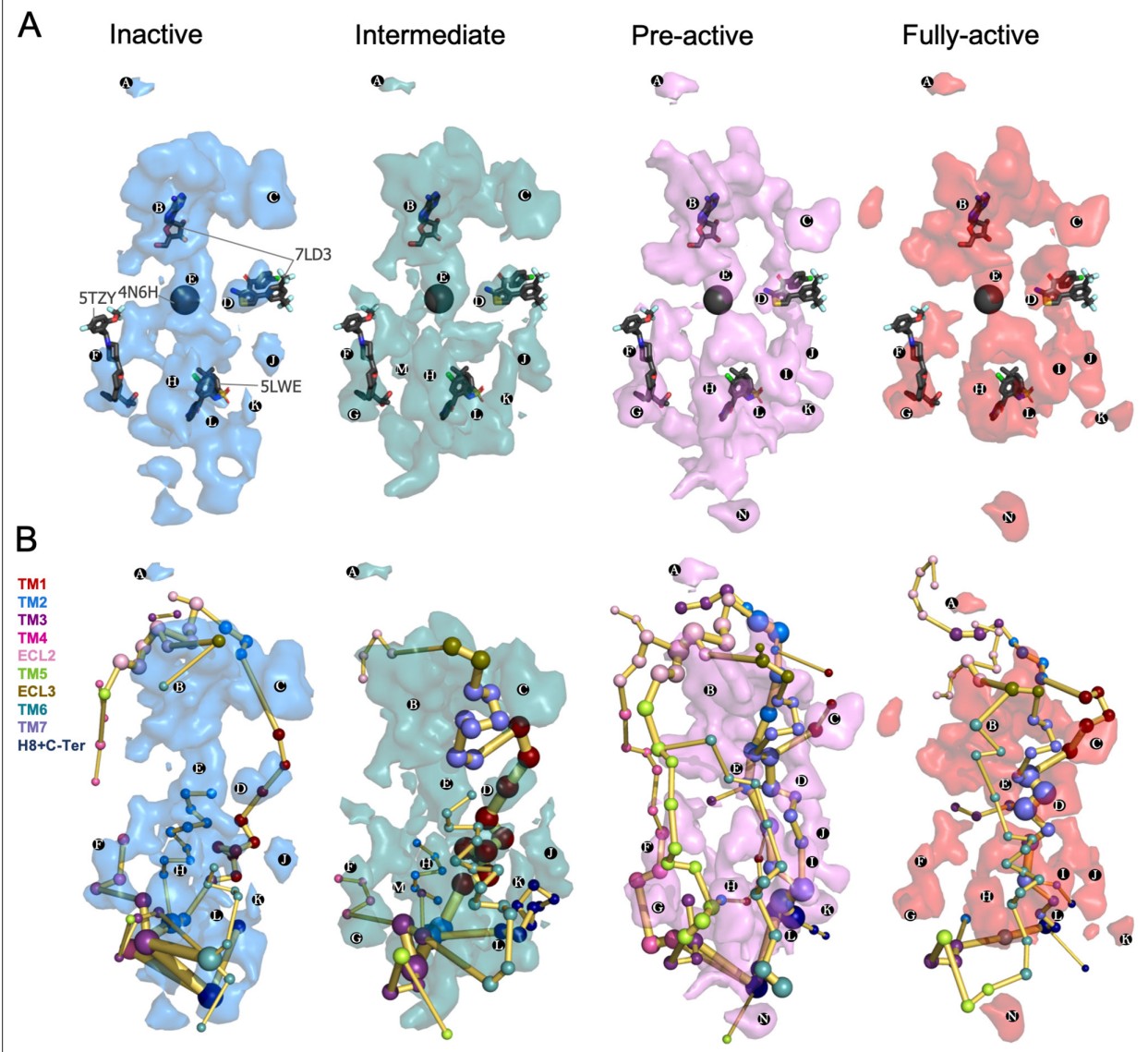

**Figure 6.** Energy coupling between the transient pockets formed along receptor activation. (**A**) Iso-surface representation of the normalized frequency map (set at $\Phi i=0.2$ iso-value) obtained from MDPocked in the inactive (blue), intermediate (teal), pre-active (violet), and fully-active (red) ensembles. The transient pockets are labeled from A to N. The allosteric modulators that overlap with the pockets found are shown as dark gray sticks (PDB 7DL3, 5TZY, and 5LWE) and spheres (PDB 4N6H). (**B**) Overlap of the transient pockets and the protein energy networks (PEN). The PEN residues (nodes) are represented by colored spheres as a function of the receptor region (e.g. TM6 nodes in teal) while the allosteric pathways (edges) as yellow-orange sticks.

complex with $A_1R$. Given its relevance for this study, **PD** is explored in more detail in the next section. We also identify a pocket (**PC**) located in an extrahelical TM1 and TM7 region. It matches with the polar region of the Monooleoylglycerol molecule captured in a GPCR structure (PDB 4MBS). **PC** is energetically coupled with TM1 and TM7 PEN residues. For a complete description of all pocket's locations and containing PEN residues, see ***Supplementary file 1***.

Regarding the lower part of the receptor, both the outer surface (extrahelical) and the inner surface (intrahelical) present many cavities. Some of them are open along the activation pathway (e.g. **PF**) and others are more predominant (e.g. **PK**) or only formed (e.g. **PG**) in some states. **PF** is formed in all states and contains extrahelical TM2-TM3-TM4 PEN residues. It matches with the AP8 allosteric modulator site found in the free fatty acid receptor 1 (PDB 5TZY)(***Lu et al., 2017***; ***Figure 6A***) and also with cholesterol and other lipid molecules found in other GPCR structures PDB 5LWE (***Oswald et al., 2016***), 4PHU (***Srivastava et al., 2014***), 5TZR (***Lu et al., 2017***), and 4XNV (***Zhang et al., 2015***).

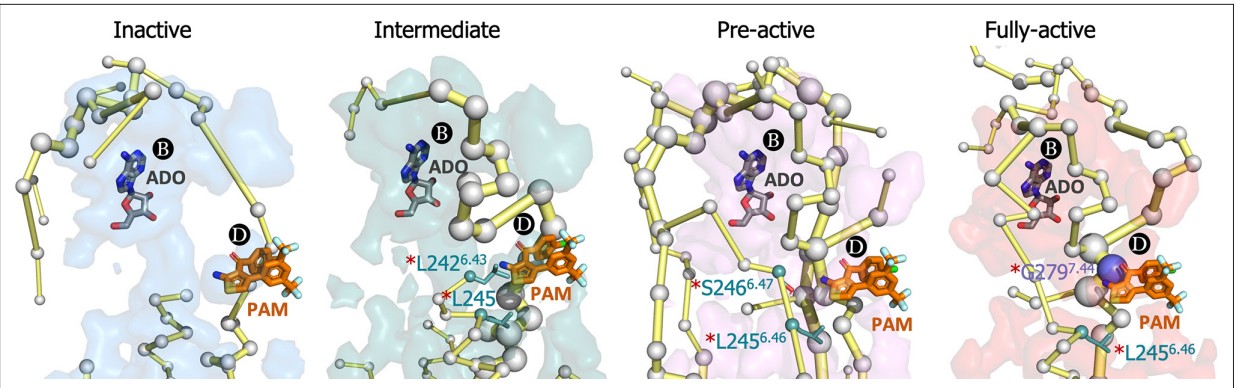

**Figure 7.** Energy coupling between pocket B and pocket D along receptor activation. Zoom view of the transient pockets and protein energy networks of the upper region of the receptor. Adenosine (ADO) in pocket B and the MIPS521 positive allosteric modulator (PAM) in pocket D, both aligned from PDB 7LD3, are depicted in gray and orange sticks, respectively. The PEN residues (nodes) are represented by light gray spheres. The size of each edge and node corresponds to their importance for allosteric communication. The experimentally identified allosteric residues located in pocked D that affect the PAM are colored as a function of the receptor region (TM6 nodes in teal and TM7 nodes in purple) and highlighted with a red asterisk. The allosteric pathways (edges) are depicted as yellow-orange sticks.

The online version of this article includes the following figure supplement(s) for figure 7:

**Figure supplement 1.** $A_1R$ shallow pocket of the MIPS521 positive allosteric modulator (PAM), PDB 7LD3.

Among GPCR allosteric modulators co-crystalized in the intracellular surface, only Vercircon (5LWE; *Oswald et al., 2016*) overlaps partially with **PL,** *Figure 6A*. This suggests that the shape and location of the $A_1R$ pockets located in the lower part are highly system-specific. In fact, the shape of the lower region changes substantially along activation. **PG** is formed from the intermediate to the fully-active state and is coupled to extrahelical TM5-TM3 PEN. **PM** supposes an interesting pocket because it is only predominant in the intermediate state. In addition, it contains many PEN residues in the extrahelical TM1-TM2-TM4 region. **PK** is more open in the intermediate and pre-active states and contains PEN residues of the extrahelical TM1-H8 region, while **PI** is only formed in the pre-active and fully-active states and is energetically coupled with extrahelical TM6-TM7 PEN. **PH** and **PN** are located in the extrahelical TM5-TM6 region. **PH** is more predominant in the pre-active and fully-active state containing many PEN residues while **PN** is only formed in the pre-active and fully active states but it is less energetically connected (*Figure 6B*). The overlap between pockets and PEN provides a view of how the distinct pockets are allosterically coupled between them and with other functional regions of the receptor (*Figure 6B* and *Supplementary file 1*).

## ADO and MIPS51 PAM have a significant impact on the energy networks

In order to establish a connection between the energy networks and the mode of action of allosteric modulators, we focus on exploring the effect of MIPS521 positive allosteric modulator (PAM) and adenosine (ADO) agonist as a proof of concept. Experimental assays and Gaussian accelerated MD determined that MIPS521 PAM increases the binding affinity of ADO in the orthosteric site (*Draper-Joyce et al., 2021*). Thus, **PB** and **PD** must be allosterically coupled. Among MIPS521 PAM pocket residues, only **L242⁶·⁴³**, **L245⁶·⁴⁶**, **S246⁶·⁴⁷**, and **G279⁷·⁴⁴** were experimentally found to affect the PAM cooperativity. Interestingly, the PEN obtained in the presence of ADO captures these key residues along activation, including TM6 (**L242⁶·⁴³** and **L245⁶·⁴⁶**) in the intermediate, **L242⁶·⁴³** and **S246⁶·⁴⁷** in the pre-active and **L242⁶·⁴³** and TM7(**G279⁷·⁴⁴**) allosteric residues in the fully-active ensemble (*Figure 7* and *Figure 7—figure supplement 1*). Indeed, **G279⁷·⁴⁴** becomes a key node in the PEN of the fully-active ensemble. This evidence suggests that although both PD and PB are open in all conformational states, their energy coupling is particularly stronger during receptor activation.

This prompted us to investigate whether the binding of ADO and MIPS521 PAM can affect the allosteric communication between PB and PD sites. To that end, we performed cMD of the heterotrimeric $G_{i2}$ protein ADO-$A_1R$-$G_{i2}$ complex in presence of the PAM (PAM-ADO-$A_1R$-$G_{i2}$ complex) and in absence of adenosine ($A_1R$-$G_{i2}$ complex) in order to compute their conformational landscape and

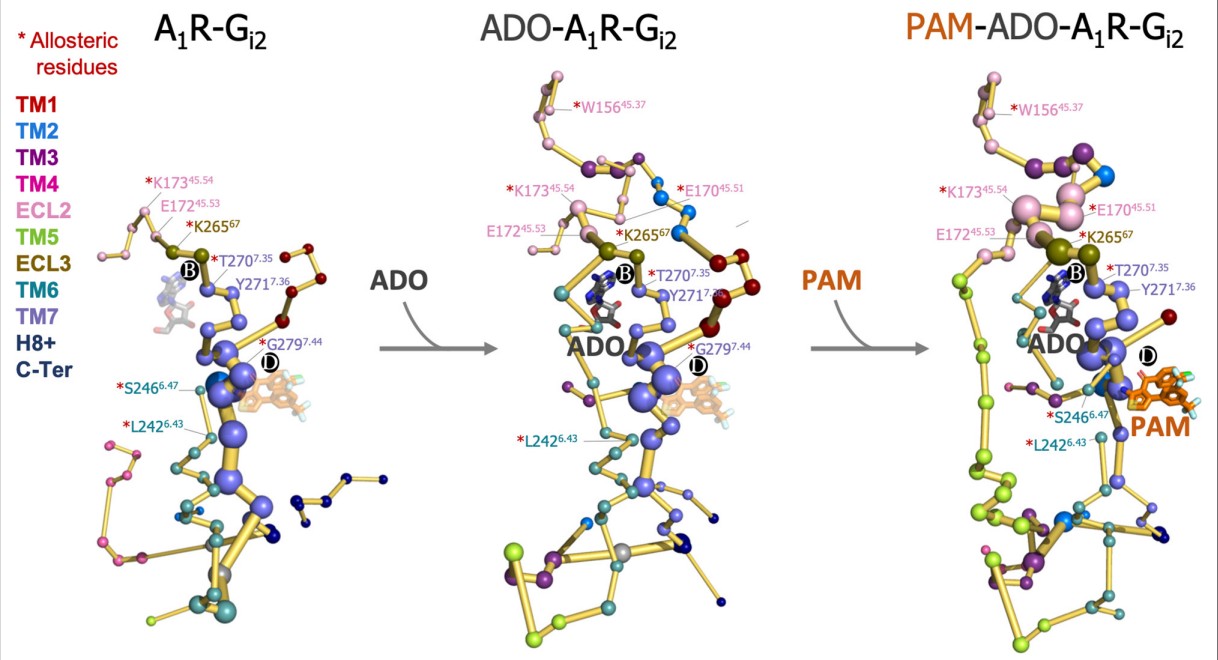

**Figure 8.** Effect of adenosine (ADO) and MIPS51 positive allosteric modulator (PAM) binding on the protein energy networks (PEN) of $A_1R$-$G_{i2}$. ADO in pocket B and MIPS521 PAM in pocket D, both aligned from PDB 7LD3, are depicted in gray and orange sticks, respectively. Note that ADO and PAM sticks are displayed with transparency in the systems in which they are absent. The PEN residues (nodes) are represented by colored spheres as a function of the receptor region (e.g. TM6 nodes in teal) while the allosteric pathways (edges) as yellow-orange sticks. The size of each edge and node corresponds to their importance for allosteric communication. The experimentally identified allosteric residues captured in the PEN nodes are labeled with a red asterisk.

The online version of this article includes the following figure supplement(s) for figure 8:

**Figure supplement 1.** Effect of ADO and MIPS51 PAM on the conformational landscape of $A_1R$ in the presence of G-protein.

energy networks following the same protocol as for the ADO-$A_1R$-$G_{i2}$ complex (*Figure 8* and *Figure 8—figure supplement 1*). The analysis of the PEN of the $A_1R$-$G_{i2}$ complex reveals that in the absence of ADO, the receptor displays a reduced allosteric communication between PB and functional regions of the receptor, such as the extracellular allosteric center, TM6, and PD allosteric site. As expected, the presence of ADO restores the allosteric coupling between PB and TM6, which could explain the increase in receptor activity associated with agonist binding. Additionally, our analysis of the PAM-ADO-$A_1R$-$G_{i2}$ complex shows that the PAM reinforces the TM7-ECL3-ECL2 allosteric pathway that couples PD with PB, and ECL2 now communicates to the intracellular region through TM5 (*Figure 8*). Notably, a recently published study reported that the orthosteric pocket (i.e. PB) contracts after ADO binding, as demonstrated by shortened distances of the so-called vestibular lid (defined as the sum of the length of the triangle perimeters formed by E170[45.51]-Y271[7.36]-E172[45.53] interacting residues) and the E172[45.53]-K265[67] salt bridge (*Li et al., 2022*). Remarkably, the TM7-ECL3-ECL2 enhanced pathway by the PAM effect contains the vestibular lid and the E172[45.53]-K265[67] salt bridge residues (*Figure 8*). This suggests that PAM promotes the contraction of PB, leading to the stabilization of the ADO-bound state. Thus, the enhanced energy coupling between PB and PD may be responsible for the increase in the binding affinity of ADO in the presence of the PAM, as observed experimentally (*Draper-Joyce et al., 2021*). This data indicates that allosteric modulators are able to enhance and redistribute the energy networks, which is likely attributed to their effects on receptor activity.

## Discussion

A comprehensive knowledge of the allosteric properties of receptors is essential for the design of allosteric drugs. It involves the deciphering of activation conformational landscapes, the decoding of allosteric networks, and the characterization of transient pockets that are allosterically coupled with

functional regions of the receptor. Here, we focus on the $A_1R$ due to the failure in the development of efficient and safe allosteric modulators reported to date (*Deb et al., 2019*; *Draper-Joyce et al., 2021*; *Nguyen et al., 2016b*).

The $A_1R$ activation FEL associated with the TM6 inward-to-outward transition reveals that the receptor is in dynamic equilibrium with inactive, intermediate, and pre-active states, where the inactive is the most stable and the intermediate and pre-active states are separated by a lower energy barrier. This fast-conformational transition observed in the presence of adenosine may favor the binding and activation of G-proteins (*Mafi et al., 2022*; *Bostock et al., 2019*). Regarding the activation pathway, the inactive state of TM6 resembles the inactive X-ray structure and progresses towards an intermediate state that is characterized by a rather high TM6 torsion and a modest TM6 opening due to the tight TM3-TM6 energy coupling in the intracellular ends. This results in torsion at the end of TM6, which is signature of the intermediate conformations. This torsion evolves to a more stable intermediate state that exhibits higher TM3-TM6 distances. Finally, TM6 progresses towards a complete opening reaching the pre-active state, in which the fully-active Cryo-EM structure is not highly populated. Interestingly a larger outward movement of TM6 is accessible, as captured in other class A GPCR receptors (*Rasmussen et al., 2011*; *Carpenter et al., 2016*). Upon G-protein coupling, the slim $G\alpha_{i2}$-α5 helix induces a population shift toward TM6 fully-active conformations that resemble the active Cryo-EM structure, and the intermediate and inactive states are not accessible. This data is in line with the combined activation mechanism described in many GPCRs by means of NMR and computational studies, which states that the combined action of both, agonist and G-proteins is essential to stabilize the adoption of a less dynamic fully-active state (*Weis and Kobilka, 2018*; *Bostock et al., 2019*; *Mattedi et al., 2020*; *Miao and McCammon, 2016*).

The conformational progression obtained from the inactive to the adoption of fully-active conformations is relevant to target $A_1R$ drug specificity, especially the hidden intermediate and pre-active states. The collection of all these conformational states is also valuable to compute the allosteric networks of the receptor along the activation pathway. Such analysis provides a dynamic view of how allosteric communication evolves along activation. The analysis of the interaction protein energy networks (PEN) captured the extra and intracellular communication centers together with the allosteric pathways that interconnect them.

Focusing on the intracellular region, a tight allosteric commutation is observed in the inactive ensemble between $D104^{3.49}$, $R108^{3.53}$, $D229^{6.30}$, and $D326^{C\text{-}Ter}$, which is key to attaining TM6 in the inactive state. The strong TM6-TM3 hydrogen network weakens in the intermediate state and $D229^{6.30}$ now communicates with $R105^{3.50}$. This transient communication prevents TM6 opening and captures the receptor in intermediate conformations, that are characterized by a structural torsion in the TM6 end. In the pre-active ensemble, TM6 opening provokes the complete loss of TM3-TM6 communication. Interestingly, the flexible $D326^{C\text{-}Ter}$ performs transient communications along the receptor activation with $R108^{3.53}$ and $K116^{4.34}$. These interactions could compensate for the TM6-TM3 break upon activation. However, $D326^{C\text{-}Ter}$ communication takes place in the region where the G-proteins bind. Thus, the binding of the $G\alpha_{i2}$-α5 helix must also compensate for the release of the C-ter from the G-proteins binding site. In fact, $G\alpha_{i2}$-α5 helix (D340) replaces the $D326^{C\text{-}Ter}$ communication with $R108^{3.53}$ becoming a hub node in the intracellular allosteric center. Regarding the allosteric pathways, we observe enhanced communication between the intra and extracellular centers upon activation. In contrast with the inactive and intermediate states, in the pre-active state, multiple energy pathways arise including TM2, TM4, TM5, TM6, and TM7. This is evidenced by the lack of TM6 communication in the inactive state, the partial TM6 communication in the intermediate state, and the complete TM6 communication in the pre-active state. Interestingly, the presence of the G-proteins fine-tunes the allosteric communication profile only passing through TM2, TM1, TM6, and TM7. These insights may be interesting for future biased agonism studies. We hypothesize that the binding of drugs that potentially favor the establishment of specific PEN profiles may result in discrimination between different intracellular partners by preferentially communicating with one of them. Thus, avoiding side effects by activating desired pathways.

Finally, we studied the connection between transient pockets and PEN along the receptor activation in order to unravel the allosteric coupling between pockets and distinct functional regions of the receptor. As a proof of concept, we focus on PD, which corresponds to the binding site of MIPS52, a positive allosteric modulator (PAM) that increases adenosine binding affinity in the orthosteric site

(PB). Although PD is open in all conformational states, the communication between PB and PD is enhanced along activation capturing the allosteric residues that were found to affect its PAM from the intermediate to the fully active. Based on this observation, we hypothesize that drugs that bind pockets and interact with PEN that progress towards regions of the receptor where the function can be altered may potentially affect the receptor activity through allosteric effects. Additionally, the pocket where the drug binds must be open at least in the conformational state that is targeted. As a practical aspect, virtual screening campaigns could use this information during the design procedure by selecting drug candidates that perform stronger interactions with PEN residues that are contained in the pockets.

To further support this hypothesis, we explored the allosteric effects of ADO and MIPS52 PAM on the PEN. Interestingly, we observed that ADO is crucial for the formation of the extracellular center and the allosteric connection between the PB site and the TM6 pathway. Furthermore, MIPS52 PAM reinforces the allosteric pathway that connects PB and PD sites and redistributes other connections. These alterations in the PEN can be related to their mode of action. ADO may increase the activity of the receptor through its communication with TM6 and the PAM may increase ADO binding affinity through stronger energy coupling between PD and PB pockets. These findings imply that the mode of action of allosteric drugs could be predicted depending on how they redistribute the PEN.

## Conclusions

With the combination of free energy landscape construction, allosteric networks decoding, and transient pockets calculation we successfully capture hidden conformational states, the allosteric communication centers/pathways, and the transient pockets formed along the $A_1R$ activation. Most importantly, by an in-depth study of the connection between these three elements, we provide a complete dynamic view of the $A_1R$ activation. Specifically, we observe that the allosteric communication is progressively enhanced between the extra and intracellular allosteric centers throughout the inactive, intermediate, and pre-active states to be fine-tuned upon the binding of G proteins in the fully-active state. In fact, not only the allosteric networks are dynamic, but the shape and frequency of formation of the transient pockets also change along the different conformational states. Overlap of the energy networks and transient pockets uncovers how the allosteric coupling between pockets and distinct regions of the receptor is altered along the receptor activation pathway. As a proof of concept, adenosine and a previously experimentally determined positive allosteric modulator were found to enhance and redistribute the energy networks of the receptor in a manner that is consistent with their respective biological activities. Understanding and predicting drug effects depending on how they redistribute the protein energy networks present a promising avenue for drug discovery. All these system-specific structural dynamics understanding provide useful information to advance the design of $A_1R$ allosteric modulators on the basis of structure-based drug design. This computational approach can be also transferable to other GPCRs and related receptors, which is of interest for the design of novel allosteric drugs.

## Materials and methods
### Conventional Molecular Dynamics (cMD) simulations
#### System preparation

In total, five systems were prepared: the inactive conformation of $A_1R$ in complex with adenosine ($A_1R$-ADO, inactive), the active conformation in complex with adenosine ($A_1R$-ADO, active), the active conformation in complex with adenosine and heterotrimeric $G_{i2}$ proteins (ADO-$A_1R$-$G_{i2}$, active), the active conformation in complex with the heterotrimeric $G_{i2}$ proteins ($A_1R$-$G_{i2}$, active) and the active conformation in complex with adenosine, the MIPS521 positive allosteric modulator and heterotrimeric $G_{i2}$ proteins (PAM-ADO-$A_1R$-$G_{i2}$, active). For the $A_1R$-ADO inactive system, we used the inactive structure of $A_1R$ in complex with PSB36 (PDB 5N2S; *Cheng et al., 2017b*) as the starting structure. PSB36 was manually removed and adenosine was placed in the orthosteric site by structural alignment with the active Cryo-EM structure of $A_1R$ (PDB 6D9H; *Draper-Joyce et al., 2018*). The $A_1R$-ADO active system was obtained by manually removing the G-proteins from PDB 6D9H. Regarding the multimeric complexes, the ADO-$A_1R$-$G_{i2}$ active system was obtained from PDB 6D9H. The crystal structure (PDB 2ODE; *Soundararajan et al., 2008*) was used to reconstruct the missing $G_{i2}$ protein regions and also

to place Guanosine-5'-Diphosphate (GDP) and the magnesium ion ($Mg^{+2}$) by structural alignment. The $A_1R$-$G_{i2}$ system was generated by manually removing adenosine from ADO-$A_1R$-$G_{i2}$ and PAM-ADO-$A_1R$-$G_{i2}$ was generated by adding the MIPS521 positive allosteric modulator to ADO-$A_1R$-$G_{i2}$ by structural alignment with PDB 7LD3. The Modeller software (*Eswar et al., 2006*) was used to reconstruct the missing X-Ray and Cryo-EM regions. MD parameters for the ligands (ADO, PAM, and GDP) were generated with the parmchk module of AMBER20 (*Case et al., 2020*) using the general amber force field (GAFF). The atomic charges (RESP model) were obtained using the antechamber module of AMBER20, with partial charges set to fit the electrostatic potential generated at HF/6–31 G* level of theory using Gaussian 09. (*Frisch et al., 2016*) Internal water molecules highly conserved among GPCRs were incorporated into the $A_1R$ internal channel of the inactive and active systems using the HomolWat web server tool. Specifically, HomolWat identified a total of 76 and 100 internal water molecules that fit into our active and inactive structures, respectively. These water molecules were incorporated from multiple PBD data. (*Mayol et al., 2020*) All systems were filled into a simulation cell composed of a phosphatidylcholine (POPC) membrane solvated at NaCl 0.15 nM using the packmol memgen tool implemented in AMBER20. The force fields selected to describe the different molecule types for the MD simulations were ff14SB (protein), GAFF2 (ligands), Lipid17 (membrane), and TIP3P (waters). In addition, two disulfide bonds were created between C80-C169 and C260-C263 residues in $A_1R$.

## Molecular dynamics protocol

The systems were minimized in a two-stage geometry optimization approach. In the first stage, a short minimization of the water molecules positions, with positional restraints on the protein, ligand, and P31 atoms of the membrane was performed with a force constant of 10 kcal $mol^{-1}$ $Å^{-2}$ at constant volume periodic boundary conditions. In the second stage, an unrestrained minimization including all atoms in the simulation cell was carried out. The minimized systems were gently heated in two phases. In the first phase, the temperature was increased from 0K to 100 K in a 40 ps step. Harmonic restraints of 10 kcal $mol^{-1}$ $Å^{-2}$ were applied to the protein, ligand, and membrane. In the second phase, the temperature was slowly increased from 100 K to the production temperature (303.15 K) in a 120 ps step. In this case, harmonic restraints of 10 kcal $mol^{-1}$ $Å^{-2}$ were applied to the protein, ligand, and P31 atoms of the membrane. The Langevin thermostat was used to control and equalize the temperature. During the heating process, the initial velocities were randomized. For the heating and following steps, bonds involving hydrogen were constrained with the SHAKE algorithm and the time step was set at 2 fs, allowing potential inhomogeneities to self-adjust. The equilibration step was performed in three stages. In the first stage, an MD simulation of 5 ns under NVT ensemble and periodic boundary conditions was performed to relax the simulation temperature. In the second stage, an MD simulation of 5 ns under NPT ensemble at a simulation pressure of 1.0 bar was performed to relax the density of the system. The semi-isotropic pressure scaling using the Monte Carlo barostat was selected to control the simulation pressure. In the third stage, an additional MD simulation of 10 ns was performed to further relax the system. After the systems were equilibrated in the NPT ensemble, we performed 3 independent MD production runs of 500 ns each (i.e. 1.5 µs accumulated time for each system). An 11 Å cutoff value was applied to Lennard-Jones and electrostatic interactions. For the PAM-ADO-$A_1R$-$G_{i2}$ system, given that the MIPS521 PAM presents low binding affinity because it mostly performs weak interactions with the receptor, we applied a slight parabolic restrain with a force constant of 10 Kcal/(mol·$Å^2$) in the distance between S246(OH) and PAM(N). This avoided unbinding of the PAM ligand during the simulation time.

## Metadynamics simulations

### Collective variables

Metadynamics is a powerful method to construct complex free energy landscapes of proteins as a function of a few low-dimensional descriptors also referred to as collective variables (CVs). (*Bussi and Laio, 2020*; *Laio and Gervasio, 2008*) In this work, we selected the TM3-TM6 intracellular ends distance as the first collective variable (CV1). Specifically, we computed the center of mass (COM) distance between the backbone atoms of TM6($R105^{3.50}$, $Y106^{3.51}$, $L107^{3.52}$, $R108^{3.53}$, and $V109^{3.54}$) and TM3($Y226^{6.27}$, $G227^{6.28}$, $K228^{6.29}$, $L230^{6.30}$, and $E230^{6.31}$). For the second collective variable (CV2), we relied on the TM6 torsion, which was described by the dihedral angle formed by the alpha carbon

atoms of TM7(T277$^{7.42}$ and T270$^{7.35}$) and TM6(L236$^{6.37}$, W247$^{6.48}$). These two CVs were used to describe and monitor the TM6 inward-to-outward transition (A$_1$R activation).

## Well-Tempered and multiple-walkers metadynamics simulation protocol

The PLUMED2.7 (*Tribello et al., 2014*) software package together with AMBER20 (*Case et al., 2020*) was used to carry out the metadynamics simulations. During a metadynamics simulation, external energy quantities (Gaussian potentials) are added at a regular number of MD steps to a selected CVs (see above). This bias potential encourages the system to escape from local energy minima and overcome energy barriers, thus allowing for enhanced sampling of the CV conformational space (*Laio and Parrinello, 2002*). After sufficient simulation time, the bias potential converges and the FEL can be reconstructed by summing the Gaussian potentials added to the CV values along the simulation time. Here, Gaussian potentials of height 0.5 kcal mol-1 and width of 0.25 (CV1) and 0.015 (CV2) were deposited every 2 ps of MD simulation at 303 K. For a smooth convergence of the bias potential, we used the well-tempered (WT)(*Barducci et al., 2008*) version of metadynamics algorithm, in which the height of the gaussian potentials were gradually decreased over time proportional to the potential deposited in the currently visited point of the CV space. A bias factor parameter of 10 was selected to control how quickly the Gaussian height is decreased. In addition, the multiple-walkers approach (*Raiteri et al., 2006*) was used to improve the conformational sampling and to speed up the metadynamics simulations. It is based on running in parallel interacting replicas (walkers) where each walker biases the identical CVs and reads the Gaussian potentials deposited by the others during the simulation, thus reconstructing the same metadynamics bias simultaneously. In particular, we run 10 walker replicas. The ten walker structures (W1-10) used as starting points for the walker metadynamics simulations were carefully selected from the initial conformational sampling of the cMD simulations. Specifically, we selected five walker structures (W1-5) from the cMD starting from the inactive X-Ray and five walker structures (W6-10) from the cMD starting from the active Cryo-EM in order to provide a path of conformations that encompasses the conformational states sampled along the A$_1$R activation pathway (*Figure 1—figure supplement 2*). Each walker replica was run for 25 ns, giving a total of 250 ns. Finally, the FEL of the TM6 inward-to-outward transition was completely reconstructed by summing the Gaussian potentials deposited by all walker replicas as a function of the CVs.

## Convergence

An indicator of convergence consists of observing that the free energy surface does not change significantly over time. We estimate the convergence of the recovered FEL for monitoring the free energy difference (ΔΔG) between the local energy minima of the activation conformational surface along the simulation time. In particular, we calculated the inactive-active, the inactive-intermediate, and the active-intermediate local energy minima differences (ΔΔG), see *Figure 1—figure supplement 3*. The metadynamics simulations were considered to be converged once we observed that with increasing simulation time, the energy differences between the energy minima tend to flatten. In other words, once the free energy surface does not change significantly during a relatively long period of time in the last part of the simulation. We have also assessed convergence by analyzing the CV1 values over simulation time. *Figure 1—figure supplement 4A* shows that during the first 100 ns, walkers primarily oscillate around their initial CV1 values. Subsequently, at around 200 ns walkers exhibit a higher frequency of crossing into regions occupied by other walkers. This is further supported by the exploration of W1 and W10, as shown in *Figure 1—figure supplement 4B*. These two walkers initially start the landscape reconstruction at the opposite extremes of the CV space. At 120 ns, they are able to escape from their respective basins and approach each other, sampling similar CV values (at approximately 240 ns). At this point of the simulation, only these two walkers have covered the entire conformational space of activation. Subsequently, they tend to return to previously sampled CV space. The observation that walkers do not become trapped in their initial CVs region, but instead explore and cross into other regions suggests that our sampling strategy, which involved starting the simulations with walkers that spanned the entire CV space of interest, has facilitated the exploration of the relevant conformational space. Although we cannot guarantee full convergence of the free energy landscape under these conditions, we successfully reconstructed the major conformational states of the receptor activation at 250 ns.

## Error

We estimated the error on the 2D free energy landscape of the first collective variable (CV1), which is the TM3-TM6 intracellular ends distance (*Figure 1—figure supplement 5*) using the block averaging technique, as described in the PLUMED tutorial on calculating error bars (https://www.plumed.org/doc-v2.8/user-doc/html/lugano-4.html). We calculated the weights using the metadynamics bias potential obtained at the end of the simulation, and assuming a constant bias during the entire course of the simulation (*Branduardi et al., 2012*). Specifically, we calculate the error using blocks of histograms of 25 ns each, covering the entire 250 ns simulation time.

## Protein energy networks (PEN) analysis

### Mean interaction energy matrix (IEM)

The get Residue Interaction eNergies and Networks (gRINN; *Serçinoğlu and Ozbek, 2018*) tool was used to calculate the pairwise residue interaction energies along the different conformational ensembles in order to obtain their respective mean interaction energy matrixes (IEM). The analysis was performed considering all residue pairs of $A_1R$, which resulted in 326x326 matrixes. For the complete $A_1R$ conformational ensemble, we used one out of two conformations sampled in the metadynamics simulations resulting in 6,250 structures. For the analysis by conformational states, we split the whole conformational ensemble obtained from metadynamics simulations (12,500 structures) into the inactive, intermediate, and pre-active ensembles, resulting in 3,648 structures for the inactive ensemble, 3,896 structures for the intermediate ensemble and 4,404 structures for the pre-active ensemble (*Figure 3—figure supplement 1*). The analysis of the multimeric complexes (ADO-$A_1$R-$G_{i2}$, $A_1$R-$G_{i2}$, and PAM-ADO-$A_1$R-$G_{i2}$) was performed using 5,000 representative structures from the cMD. For this case, we consider both the intra ($A_1$R-$A_1$R) and inter ($A_1$R-$G\alpha_{i2}$) residue pairs of interactions. In all matrixes, the positive energies (repulsive interactions) were set to 0 and the negative energies (attractive interactions) were converted to absolute values. In the second step, the matrixes were normalized. Hence, the attractive interactions were weighted containing values that range between 0 and 1.

### Shortest Path Map

To study the allosteric communication by means of PEN, we processed the normalized mean IEM into to Shortest Path Map (SPM) tool (*Maria-Solano et al., 2021*; *Romero-Rivera et al., 2017*; *Osuna, 2021*). The SPM algorithm first constructs a network graph in which only the pair of residues with a normalized mean interaction energy (IE) higher than 0.1 are considered nodes. The length of the edge connecting the pair of residues (nodes) is drawn according to their normalized mean IE value ($d_{ij}$=-log $|IE_{ij}|$). Thus, higher normalized mean IE values (closer to 1) will have shorter edge distances, whereas lower normalized mean IE values (closer to 0.1) will have edges with long distances. At this point, we apply the Dijkstra algorithm to identify the shortest path lengths and generate the SPM graph. The Dijkstra algorithm operates through all nodes of the initial network graph and determines the shortest path to go from one node origin to all other nodes. The exploration is over when all nodes have been targeted as origin. In the SPM graph, the width of each edge and the size of each node are proportional to the number of shortest paths passing through that edge or node during the exploration. The method, therefore, offers the visualization of which nodes and edges are more frequently used for going through all residues of the protein, that is they are more central and significant for the communication pathway.

## Transient pocket analysis

The MDpocket (*Schmidtke et al., 2011*) tool was used to detect transient pockets in the inactive, intermediate, pre-active, and fully-active ensembles. MDpocket is a fast geometry-based algorithm that relies on the concept of alpha spheres and makes extensive use of Voronoi tessellation during cavity detection. The output provides a normalized frequency map allowing for an iso-surface representation. In the frequency map, the iso-values ($\Phi i$) range from 0 to 1, allowing for visualization of both permanent ($\Phi i=1$) and transient pockets ($0<\Phi i<1$). Since we were interested in the detection of transient pockets along the receptor activation we used a $\Phi i=0.2$ iso-value for the display of pockets in all conformational ensembles studied.

## Acknowledgements

This work was supported by the Bio & Medical Technology Development Program (NRF-2022M3E5F3080873), the Mid-career Researcher Program (NRF-2020R1A2C2101636), the Medical Research Center (MRC) grant (NRF-2018R1A5A2025286), and the Brain Pool Program (NRF-2021H1D3A2A02038434) funded by the Ministry of Science and ICT (MSIT) through the National Research Foundation of Korea (NRF). It was also supported by the Ewha Womans University Research Grant of 2021. We are grateful to the Korea Institute of Science and Technology Information (KISTI) Supercomputing Center for providing computing resources and technical assistance (KSC-2022-CRE-0517). We thank Prof. Ferran Feixas, Dr. Raudah Lazim, and Dr. Maninder Singh for the helpful discussions and comments on the manuscript.

## Additional information

### Funding

| Funder | Grant reference number | Author |
| --- | --- | --- |
| Ministry of Science and ICT, South Korea | NRF-2022M3E5F3080873 | Sun Choi |
| Ministry of Science and ICT, South Korea | NRF-2020R1A2C2101636 | Sun Choi |
| Ministry of Science and ICT, South Korea | NRF-2018R1A5A2025286 | Sun Choi |
| Ministry of Science and ICT, South Korea | NRF-2021H1D3A2A02038434 | Miguel A Maria-Solano Sun Choi |

The funders had no role in study design, data collection and interpretation, or the decision to submit the work for publication.

### Author contributions

Miguel A Maria-Solano, Conceptualization, Data curation, Formal analysis, Investigation, Visualization, Writing – original draft; Sun Choi, Resources, Supervision, Funding acquisition, Writing – review and editing

### Author ORCIDs

Miguel A Maria-Solano (iD) https://orcid.org/0000-0002-7837-0429
Sun Choi (iD) http://orcid.org/0000-0002-7669-7954

Joint Public Review: https://doi.org/10.7554/eLife.90773.2.sa1

## Additional files

### Supplementary files

• Supplementary file 1. Transient pockets location and containing PEN residues in all conformational states.

• MDAR checklist

### Data availability

All the Mean Interaction energy matrixes (IEMs) computed have been deposited on GitHub (https://github.com/biochem0904/A1R-ENetDyna copy archived at *Biochem0904, 2023*), which will allow users to perform customized analysis of the protein energy networks. We have also deposited the PyMOL sessions corresponding to Figure 6 for a more comprehensive visualization and analysis.

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
