## [Editor Report · eLife assessment]

The authors describe the dynamics underlying allostery of the adenosine A1 receptor, providing **valuable** insights into the receptor's activation pathway. The enhanced sampling molecular dynamics simulations of available structural data, followed by network analysis, reveal transient conformational states and communication between functional regions. The authors carefully state the limitations of their work, including the restricted convergence of the free energy landscape and missing water-mediated hydrogen bond coordination. Collectively, they provide a **convincing** framework for advancing rational design strategies of specific modulators with desired modes of action.

[Editors' note: this was originally reviewed and assessed by Biophysics Colab]

---

## [Referee Report · Joint Public Review]

The objectives of the study:

This paper aims to characterize the dynamics that drive allostery of the adenosine A1 receptor (A1R) via computational analysis of its activation free energy landscape and measurements of the appropriate geometrical parameters. This is done by focusing on the allosteric signaling pathways in different activation states, from inactive to active states via intermediate and pre-active ones, as well as the characterization of putative drug-binding pockets. The long-term objectives are to eventually be able to aid drug discovery efforts for this therapeutically important

GPCR.

Key findings and major conclusions:

Conventional MD does not enable the sampling of the complete conformational landscape of receptor activation. Instead, enhanced sampling MD simulations are required to achieve this. Using metadynamics, the authors decipher the activation pathway of A1R, decode the allosteric networks and identify transient pockets. The protein energy networks computed throughout the inactive, intermediate active, pre-active and active conformational states unravel the extra and intracellular allosteric centers and the communication pathways that couple them, whereby the pathways are reinforced in the activated state. These conformations primarily differ in the dynamics of the ionic lock motif that couples TM3 to TM6 in the inactive conformation and reveal that G-proteins are required to fully stabilize the active conformation. Support for these findings comes from prior mutagenesis work on the A1R that identified key allosteric residues that in many cases map to identified communication nodes. Finally, the authors identified allosteric pockets throughout the A1R in four different conformational states that support prior experimental and MD studies on the mechanism of the positive allosteric modulator MIPS521 and which could be targeted for the design of new modulators. This indicates how energy networks are enhanced and redistributed by allosteric modulators and how this might explain their effect on receptor activity. Overall, these findings provide complementary support to a structure-based mechanism of activation and allosteric modulation of A1R, and extend the findings to incorporate dynamics across the full activation pathway.

The perceived strengths and weaknesses:

This preprint employs a combination of computational techniques to successfully reconstruct and analyze the conformational ensemble of the A1R activation. The metadynamics simulations supported the aim of the study, the results are clearly presented, and the work is very well written. The authors provide a valuable discussion of how the protein energy network analysis can contribute to the rational design of specific A1R modulators with desired mode of action. The employed computational approach does not capture communication pathways that involve water-mediated connections or interactions between ligands and residues. Moreover, full convergence of the free energy landscapes is not guaranteed. Overall, A1R is a good choice as the target for this study as there is existing structural and pharmacological data to support preliminary findings. Moreover, the framework presented herein could be adapted and scaled to other GPCRs with structural templates, which might enable comparison of allosteric pathways across families and classes.